# Experimental Investigation of Load-Bearing Capacity in EN AW-2024-T3 Aluminum Alloy Sheets Strengthened by SPIF-Fabricated Stiffening Rib

**DOI:** 10.3390/ma17081730

**Published:** 2024-04-10

**Authors:** Hassanein I. Khalaf, Raheem Al-Sabur, Andrzej Kubit, Łukasz Święch, Krzysztof Żaba, Vit Novák

**Affiliations:** 1Mechanical Department, Engineering College, University of Basrah, Basrah 61004, Iraq; hassanein.khalaf@uobasrah.edu.iq; 2Department of Manufacturing and Production Engineering, Faculty of Mechanical Engineering and Aeronautics, Rzeszow University of Technology, 35-959 Rzeszów, Poland; akubit@prz.edu.pl; 3Department of Aerospace Engineering, Rzeszow University of Technology, 35-959 Rzeszów, Poland; lukasz.swiech@prz.edu.pl; 4Department of Non-Ferrous Metals, AGH University of Science and Technology, 30-059 Krakow, Poland; krzyzaba@agh.edu.pl; 5Department of Manufacturing Technology, Faculty of Mechanical Engineering, Czech Technical University in Prague, Technicka 4, 166 07 Prague 6, Czech Republic; vit.novak@fs.cvut.cz

**Keywords:** stiffening ribs, single-point incremental forming (SPIF), digital image correlation (DIC), deformation, load-bearing capacity

## Abstract

The aluminum strength-to-weight ratio has become a highly significant factor in industrial applications. Placing stiffening ribs along the surface can significantly improve the panel’s resistance to bending and compression in aluminum alloys. This study used single-point incremental forming (SPIF) to fabricate stiffening ribs for 1 mm and 3 mm thick aluminum alloy EN AW-2024-T3 sheets. A universal compression machine was used to investigate sheet deformation. The resulting deformation was examined using non-contact digital image correlation (DIC) based on several high-resolution cameras. The results showed that deformation progressively escalated from the edges toward the center, and the highest buckling values were confined within the non-strengthened area. Specimens with a larger thickness (3 mm) showed better effectiveness against buckling and bending for each applied load: 8 kN or 10 kN. Additionally, the displacement from the sheet surface decreased by 60% for sheets 3 mm thick and by half for sheets 1 mm thick, which indicated that thicker sheets could resist deformation better.

## 1. Introduction

Over the past 20 years, the strength-to-weight ratio has become a highly significant factor in engineering [1,2]. Among the aluminum alloys, which are highly sought after in various industries, 2024-T3 stands out as an ideal subject for research in contexts where strength, durability, and lightness are key considerations [3]. The relative presence of copper followed by magnesium in the EN AW-2024-T3 [4] aluminum alloy gives it the strength and hardness, which characterize it, especially in pressure-bearing applications, despite their effect on corrosion resistance and weldability challenges [5,6].

In engineering applications, stiffening ribs stand out as structural strengthening elements; they are often added to metal panels and sheets and have recently been extended to include composite materials and polymers to enhance their rigidity, strength, and resistance to deformation under load. Usually, stiffening ribs are raised or protruding features, which run along the surface of the sheets.

Depending on their shape, many types of stiffening ribs are available, such as straight ribs, curved ribs, perforated ribs, grid ribs, tapered ribs, embossed ribs, and composite ribs [7]. In general, using stiffening ribs is a crucial part of improving the panel’s structural integrity [8]. The strategic placement of ribs along the surface can significantly improve the panel’s resistance to bending and compression [9]. The reinforcement technique increases the load-bearing capacity and mitigates the risk of buckling and deformation under applied loads [10]. Stiffening ribs have been used as a strengthening element in ferrous [11] and non-ferrous materials [12], and they have also been extended to include polymers and composites [13].

Stamping in a punch–die system is the most widely used traditional forming method for stiffening ribs. Furthermore, the single-point incremental forming (SPIF) method has also emerged in recent years [14,15]. It is worth noting that SPIF can be symmetric or asymmetric, which is determined by the nature of the desired shaping process applications. The first type is preferred for all simple geometric shapes with better distribution of stresses, while the other type (often called asymmetric incremental sheet forming (AISF)) can form more complex shapes but is more wasteful of materials [16]. Furthermore, the SPIF is a versatile manufacturing process, which offers numerous advantages in producing complex geometries with minimal tooling requirements. SPIF operates by incrementally deforming the material using a single-point tool, allowing precise control over the shaping process [17]. SPIF involves a backing plate, a blank holder, and a rotating tool mounted on a CNC machine for single-point forming on a sheet metal blank, as depicted in Figure 1. In the process of SPIF, the sheet is secured at its edges. At the same time, a tool—usually shaped like a sphere or a cone—applies pressure to the material’s surface in specific regions, gradually shaping the material to the desired form. This tool follows a pre-determined path, moving in small increments to displace the material and enabling accurate control over the shaping process. As the tool moves, the material undergoes plastic deformation, stretching, and bending to conform to the tool’s shape. This method makes fabricating stiffening ribs straight onto aluminum panels possible, providing an economical and practical way to improve their mechanical characteristics.

In recent years, many institutions have funded research into the manufacture of stiffening ribs using SPIF, which has resulted in many valuable studies. Trzepieciński et al. [18] aimed to inspire and guide researchers by delineating the ongoing developmental trends in the valuable contributions made in SPIF for lightweight metallic materials, especially SPIF process conditions and contact effects. Kubit et al. [19] analyzed the possibility of forming stiffening ribs in composite materials using the SPIF method. They succeeded in fabricating the required stiffening ribs in a unique composite material called LITECOR^®^ and determining the maximum depth of the embossing. Additional studies analyzing the LITECOR^®^ composite panels were later introduced. Raheem Al-Sabur et al. [20] analyzed the roughness and effects on the surface texture of SPIF-strengthened LITECOR^®^ composite panels. Their study investigated the influences of tool rotational speed and feed rate, revealing that the highest feed rate and tool rotational speed resulted in the smoothest surface texture with the lowest maximum height value. Another study focused on residual stresses resulting from the creation of stiffening ribs in composite materials formed by SPIF. They observed that the residual stress values in both directions were inversely proportional to the thickness of the core of the composite material [21]. Trzepieciński et al. [22] focused on the surface finish analysis of EN AW-7075-T6 and EN AW-2024-T3 aluminum alloy panel stiffening ribs during the SPIF process. They discovered that increasing the incremental vertical step size led to more pronounced ridges on the inner surface of stiffened ribs, particularly noticeable in Alclad aluminum alloy sheets. Furthermore, they reinforced their findings by employing an artificial neural network (ANN) model. Sherwan M. Najm et al. [23] studied the hardness variation of the AA1100 aluminum alloys strengthened by SPIF, and based on the research parameters, different regression equations were generated to calculate hardness. The study was also supported by ANN modeling. Habeeb et al. [24] examined the effect of step depth, sheet thickness, and wall angle influences on the surface roughness of stiffening ribs fabricated by SPIF in Al 2024-O draw pieces of an aluminum alloy by applying an analysis of variance (ANOVA) based on a full factorial design. Ján Slota et al. [25] examined local buckling in an aluminum alloy, which was stretched by the SPIF process. They found that two types of buckling can occur: (i) initial destabilization of the rib occurred halfway up its height, resulting in breakage, followed by deflection of the panel opposite the rib’s position; and (ii) the panels experienced buckling toward the rib midway up the panel height, with no significant loss of rib stability. Moreover, they compared the load–displacement curves of strengthened and non-strengthened 2xxx and 7xxxx aluminum sheets. They concluded that the rib destabilized at half its height, leading to breakage, followed by the panel deflecting in the opposite direction of the rib’s position. Andrzej Kubit et al. [26] conducted a strength analysis of SPIF-strengthened thin-walled composite material comprising reinforced epoxy and glass laminate aluminum. The study included several tests, such as tensile/shear, peel drum, uniaxial tensile, and bending tests, in a wide range of temperatures (−60 °C, room temperature, and +80 °C). The study revealed that temperatures did not impact either tensile or shear strength. 

In recent years, digital image correlation (DIC) has begun to dominate scientific research. Its uses have expanded to competing with traditional methods in measuring displacement [27], strain [28], deformation [29], and surface pattern [30]. The tested surface in the digital image correlation is divided into a set of facets of white (100%) and black (0%) according to their pixel size, which ranges between 10 and 30 pixels, which allows tracking changes in the location of any point on the recorded surface and creating maps of distortions or displacements [31]. 

It is evident that there is a continued need for additional research in the field of SPIF-strengthened materials, particularly aluminum alloys, such as EN AW-2024-T3 aluminum sheets. The primary objective of the current study is to address this gap and elucidate the axial-load shortening curves in EN AW-2024-T3 aluminum sheets strengthened by SPIF. Additionally, the study analyzes the deformation behavior of the vertical section under various compressive loads. It also assesses how load-bearing capacity affects the thicknesses of EN AW-2024-T3 aluminum alloy sheets and investigates the deformation effect on the strengthened aluminum alloy sheets using an advanced digital image correlation (DIC) system. In summary, the study aims to integrate these comprehensive analyses to develop a new vision for understanding the behavior of SPIF-strengthened aluminum alloys.

## 2. Materials and Methods

In this study, the SPIF process was used to obtain the required stiffening ribs in the aluminum alloy EN AW-2024-T3. The TM-1P Toolroom Mill (Hass Automation, Oxnard, CA, USA) provides precise control through the Haas CNC system—a necessity for executing the SPIF process. Two thicknesses (1 and 3 mm) of Alclad EN AW-2024-T3 aluminum alloy sheets (Kaiser Aluminum, Spokane, WA, USA) were investigated to produce the required stiffening ribs. The chemical composition of 2024-T3 aluminum alloy sheets includes copper as a primary alloying element, followed by magnesium, with a spectrum of other elements, as shown in Table 1.

A Z100 universal testing machine (Zwick/Roell, Ulm, Germany) was employed to determine the mechanical properties of EN AW-2024-T3 aluminum alloy sheets at room temperature, following the ISO 6892-1 standard [33]. The resulting mechanical properties are shown in Table 2.

Aluminum alloy EN AW-2024-T3 sheets with 160 mm length, 100 mm width, and thicknesses of 1 mm and 3 mm, respectively, were used. The SPIF-strengthening process was applied to a 110 mm long and 20 mm wide piece, while the forming depth was 5 mm. Figure 2a–c shows the sheet dimensions used, the TM-1P Toolroom Mill, and the sheet’s shape after the SPIF process’s completion, respectively. A steel pin—specifically an HS2-9-2 (1.3348) variant featuring a rounded tip with a radius of 3.5 mm—was employed for the forming process. The SPIF process requires reducing the friction rates between the aluminum sheets and the HS2-9-2 tool (1.3348) to obtain accurate dimensions and avoid unwanted deformations; therefore, high-quality oil Getriebeoel SAE 75W-85 (Mannol, Wedel, Germany) was used.

When the aluminum alloy EN AW-2024-T3 sheets were rib-stiffened, both the 1 mm thickness and the 3 mm thickness were examined in a compression test using the Z100 testing machine (Zwick/Roell, Ulm, Germany) for two ultimate loads of 8 kN and 10 kN, respectively. Using a Z100 machine based on the ASTM E9/E9M standard, all EN AW-2024-T3 aluminum samples strengthened by SPIF were compressed using two compressive loads (8 kN and 10 kN) at room temperature. The compression testing machine securely mounted the specimens, aligned them perpendicular to the loading direction, and fixed them well to prevent slippage during the test. Then, the compressive force was applied to the specimen at the test speed (or initial strain rate of 1 mm/min) until failure occurred or until the desired deformation was reached, as shown in Figure 3.

Furthermore, a high-resolution 2D non-contact digital image correlation ARAMIS system (GOM GmbH, Braunschweig, Germany) was used to investigate the deformation of aluminum alloy EN AW-2024-T3 sheets due to the compression test. The basic principle of the ARAMIS system is based on the digital image correlation (DIC) of several cameras projected onto the surface of the specimens [34]. Furthermore, ARAMIS analyzes facial distortion in the XY plane while measuring the change in height along the Z-axis (closer to or farther from the cameras) [35]. Figure 4 indicates the main ARAMIS cameras and their system map of vertical displacements during the compression tests of 10 kN and 8 kN. More details regarding these tests will be discussed in the next section.

## 3. Results and Discussion

### 3.1. Axial-Load Shortening Curve

The load versus end-shortening curves characterizes the average stress–strain response of longitudinally stiffened panels subjected to compressive loading [36]. The axial-load shortening curves provide a valuable understanding of the material’s deformation characteristics, elucidating how it responds to varying levels of applied loads [37]. Usually, the X-axis represents the axial displacement or shortening of the material. It represents the deformation or compression experienced by the material under the applied axial load during compression testing, represented by the Y-axis. In general, the axial-load shortening curves serve as informative tools for assessing axial stiffness, yield characteristics, and failure mechanisms in sheets while also facilitating the observation of local buckling behavior in aluminum sheets.

In aluminum applications, particularly in the aerospace industry, understanding the behavior of SPIF-strengthened aluminum sheets toward different compressive loads is crucial to predicting potential failure and structural integrity.

First, a compression machine with a measuring head ranging from 0 to 10 kN was selected for both SPIF-strengthened aluminum alloy EN AW-2024-T3 sheet specimens to examine the load capacity and local buckling behavior and improve measurement accuracy. However, the samples did not buckle during the first attempt and exceeded this range; therefore, the measuring head was replaced with a more extensive range, and the test was repeated. The applied load was gradually increased, and the specimens’ corresponding deformation (shortening) was measured, as shown in Figure 5. For the first SPIF-strengthened aluminum alloy EN AW-2024-T3 specimen with 1 mm thickness (P1), as the compressive load was applied, the specimen underwent elastic deformation, where the shortening displacement increased linearly with the applied force, representing the elastic region. Beyond the elastic region, the shortening displacement continued to increase with increasing force, but the rate of increase slowed down. It was noticed that this specimen reached the plastic region, but it did not fail, and no buckling occurred despite reaching the maximum allowable force of the compression test machine. Then, the head of the compression test machine was increased. The same previous sample, which had prior plastic deformation without a dent (P1), was again subjected to a compression test after increasing the head of the compression test machine. This retested sample is referred to as P1-1. Specimen P2 (3 mm thick) had a larger cross-sectional area than P1. As a result, P2 experienced less stress (force per area) than P1 for the same magnitude of the compressive load range. Furthermore, P2 (higher cross-sectional area) had less shortening displacement than P1-1 (smaller cross-sectional area), which matched with the results from Refs [38,39] despite not appearing clearly in Figure 4. However, this confusion can be eliminated if it is remembered that sample P1-1 is a sample in which a prior deformation occurred in the previous test. At the end of compression tests, the axial-load shortening curve of samples P1-1 and P2 sharply decreased, indicating the specimens’ failure or complete buckling.

### 3.2. Deformation of SPIF-Strengthened Ribs

As previously mentioned, aluminum alloy EN AW-2024-T3 sheets measuring 160 × 100 mm were strengthened using SPIF; the first category of sheets was 1 mm thick, while the other was 3 mm thick. The sheet part, which underwent strengthening, was centered at a length of 110 mm, and the maximum concavity of the specimens after the SPIF process was 5 mm. In the compression test, two types of loads were applied to each type of specimen, with the first having a maximum force of 8 kN and the second having a maximum force of 10 kN. This section highlights the nature of the deformations occurring in the stiffened ribs for each type of specimen depending on the thickness of the sample. The results were obtained using the ARAMIS system based on the digital image correlation (DIC) of the applied cameras.

For specimens 1 mm thick, it can be seen that these specimens did not undergo significant deformation when a compressive load of 8 kN was applied. Figure 6a shows that the deformation was limited, especially from the center of the strengthened area, especially from the edges, where it was less than 0.4 mm. It gradually increased from the edges toward the center, and the buckling reached approximately 0.8 mm. The highest buckling values were not recorded in the strengthened area compared to the non-strengthened area, where buckling reached 4 mm (blue region) at the side edges of the non-strengthened area. When the applied load was increased to 10 kN, as in Figure 6b, the deformation area (blue area) expanded more clearly, especially in the non-strengthened area, compared to the previous case. However, the strengthened area continued to provide good resistance to buckling and was not significantly affected, except under a significant increase in strength.

Specimens with a larger thickness (3 mm) showed better effectiveness against the applied load, whether 8 kN or 10 kN, as shown in Figure 7a,b. In both cases, the areas with more significant deformation (blue areas) were reduced compared to samples with a thickness of 1 mm. The resultant behavior can be explained by the larger cross-sectional area of the thicker specimens, which allows them to withstand higher loads before reaching the point of yield or failure, resulting in enhanced load-bearing capacity [40]. Moreover, thicker specimens are less susceptible to buckling, as higher thickness provides excellent resistance against buckling, reducing deformation areas compared to thinner specimens [41].

Figure 8 shows the pictures of samples taken during the test period for the four cases used in this study. In the samples with a thickness of 1 mm, buckling was apparent when the load was 8 kN (Figure 8a), and when the load was increased to 10 kN, failure occurred in the middle of the sample (Figure 8b). In the samples with a larger thickness (3 mm), buckling was lower than in the previous samples with a thickness of 1 mm (Figure 8c). It can also be noted that despite the increase in the applied load to 10 kN, failure did not occur despite the beginning of deformation in the middle of the sample (Figure 8d), which strengthens the previous result, where the strengthening of aluminum alloy EN AW-2024-T3 sheets using SPIF gave positive results and was better in the samples with greater thickness.

Figure 9 shows deformation occurring in 1 mm thick aluminum specimen EN AW-2024-T3 sheets just after destruction. It shows that a large deformation occurred, especially in approximately the middle of the sample, whether in the SPIF-strengthened area or other areas, until it reached about 30 mm, where the largest buckling (partial destruction) occurred (blue color). It can also be noted that the deformation gradually decreases as we move vertically away from the middle of the sample. There are several reasons to which this behavior can be attributed, such as stress concentration, which causes localized deformation and eventual buckling at the weakest area of the sheet [42]. Moreover, the compression test in thin sheets is accompanied by bending. The highest bending moment occurs at the center of the sheet, leading to maximum deformation and buckling in this region [43], in contrast to the edges, where the edges of the sheet provide some degree of support and constraint against buckling compared to the center [44].

### 3.3. Deformation along Section Length

This part continues the study of the cases of deformation occurring in aluminum 2024-T3 sheets, specifically along the part receiving strengthening, which is 110 mm long and 5 mm wide. Figure 10 focuses on sheets 1 mm thick, while Figure 11 depicts samples 3 mm thick. In both forms, the test was carried out under a compressive load of 8 kN and another of 10 kN, and the two cases were compared with the no-load condition (0 kN). The measurements (Z component) on the vertical axis represent the coordinates of points along the length of the groove (stiffening rib) relative to the sheet surface (distance from the sheet surface). In all cases, the two ends of the stiffening rib had no effect due to their proximity to the support areas, and they were also the least concave areas. The resulting behavior can be seen along the first 10 mm of the specimen and the last 10 mm close to the second end. Concavity occurs in the stiffening rib after moving away from the ended areas.

Figure 9 shows that in the absence of any load (*p* = 0 kN), there was no displacement and no dent in specimens with a thickness of 1 mm. The maximum displacement from the sheet surface is approximately 3.25 mm at the middle of the length of the stiffening rib. When a pressure load of 8 kN is applied, bending begins, and the distance from the sheet surface decreases by approximately 18%. When a load greater than 10 kN is applied, a more considerable dent occurs, and the distance from the sheet surface also decreases until it reaches approximately 30% compared to the no-load case. As expected, Figure 10 shows that specimens with a higher thickness rendered higher displacement values from the sheet surface, reaching approximately 3.5 mm in the unloaded state. With loading, the two cases showed a decreasing displacement from the sheet surface until reaching approximately 60% at a distance of 65 mm from the lower end. This increase in shrinkage reflects the ability of aluminum alloy EN AW-2024-T3 sheets reinforced with stiffening ribs to stack more and thus have a higher ability to resist buckling, especially for samples with higher thicknesses.

## 4. Conclusions

In this study, aluminum alloy EN AW-2024-T3 sheets measuring 160 × 100 mm were strengthened using SPIF; the first category of sheets was 1 mm thick, while the other was 3 mm thick. Two types of compressed loads were applied to each type of specimen: 8 kN and 10 kN. The deformations occurring in the stiffening ribs were examined using digital image correlation (DIC). Several conclusions can be drawn from this study, as follows:The deformation gradually increased from the edges toward the center, and the highest buckling values were not recorded in the non-strengthened area.The specimens did not undergo significant deformation under a compressive load of 8 kN, which was limited, especially from the center of the strengthened area and the edges.Specimens with a larger thickness (3 mm) showed better effectiveness against buckling and bending under each applied load: 8 kN or 10 kN.Aluminum alloy EN AW-2024-T3 sheets reinforced with stiffening ribs could stack more, making them less likely to buckle, especially with thicker samples.During deformation resulting from the compression tests, the parts of the stiffening ribs converged. Thus, the displacement from the sheet surface decreased by 60% for sheets 3 mm thick and half of this ratio for sheets 1 mm thick, which reflected greater strength toward the resulting deformations.

## Figures and Tables

**Figure 1 materials-17-01730-f001:**
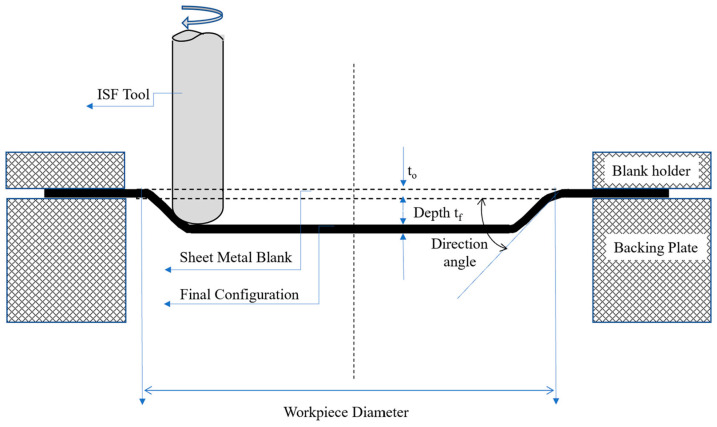
Schematic representation of symmetric SPIF.

**Figure 2 materials-17-01730-f002:**
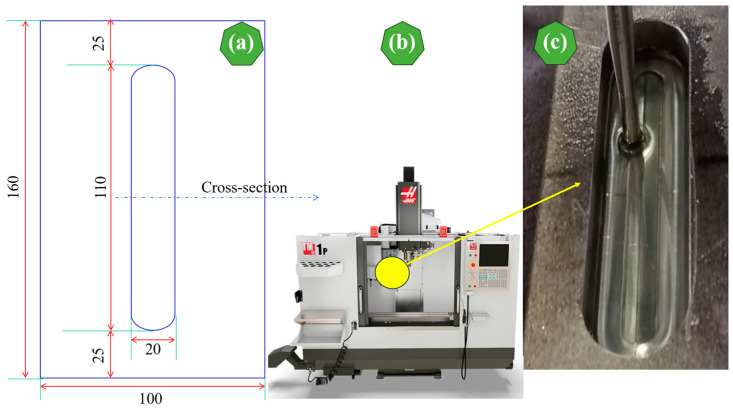
(**a**) Dimensions (in mm) of a SPIF stiffened rib, (**b**) TM-1P milling machine, and (**c**) Aluminum alloy EN AW-2024-T3 sheets after SPIF process completion.

**Figure 3 materials-17-01730-f003:**
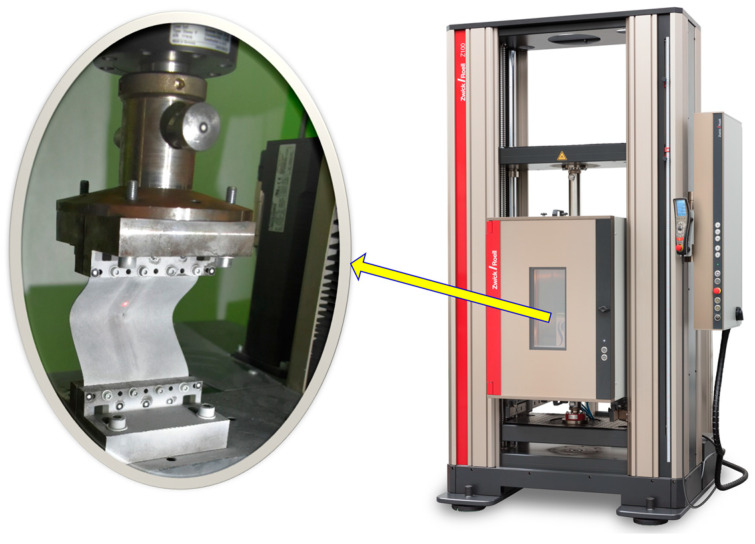
A compression machine with specimen fixation (after test).

**Figure 4 materials-17-01730-f004:**
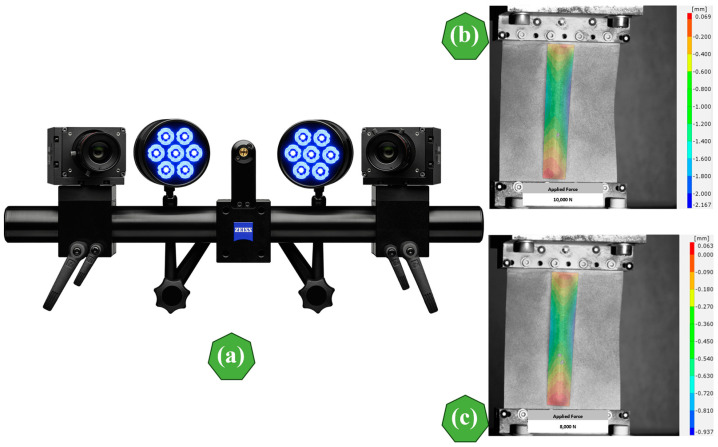
ARAMIS system: (**a**) Main cameras and project lights, and system map of specimen deformation under (**b**) compression test using 10 kN and (**c**) compression test using 8 kN.

**Figure 5 materials-17-01730-f005:**
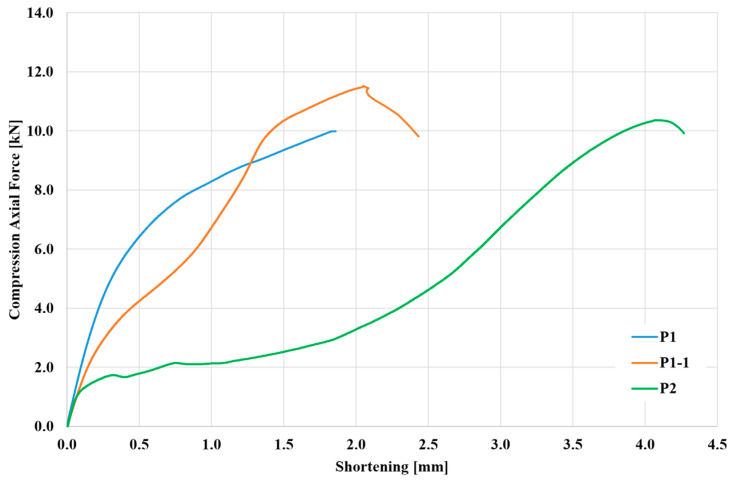
Shortening of samples with 1 mm thickness (P1 and P1-1) and 3 mm thickness (P2).

**Figure 6 materials-17-01730-f006:**
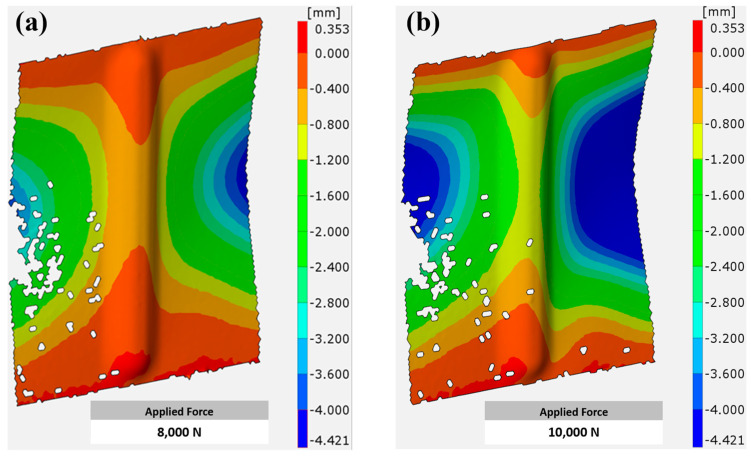
Deformation of 1 mm thick aluminum alloy EN AW-2024-T3 sheets under compression load of (**a**) 8 kN, (**b**) 10 kN.

**Figure 7 materials-17-01730-f007:**
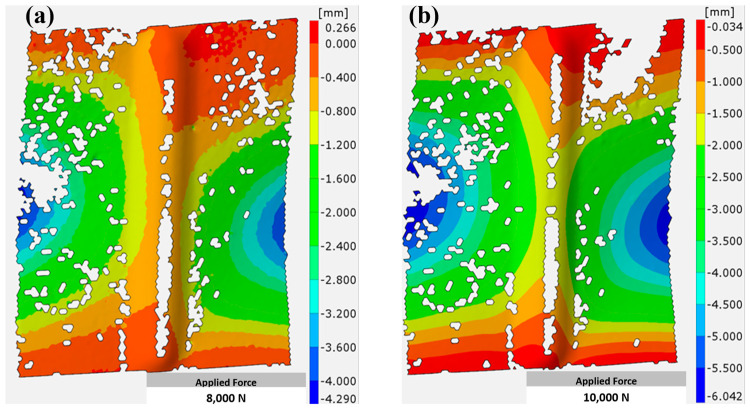
Deformation of 3 mm thick aluminum alloy EN AW-2024-T3 sheets under compression load of (**a**) 8 kN, (**b**) 10 kN.

**Figure 8 materials-17-01730-f008:**
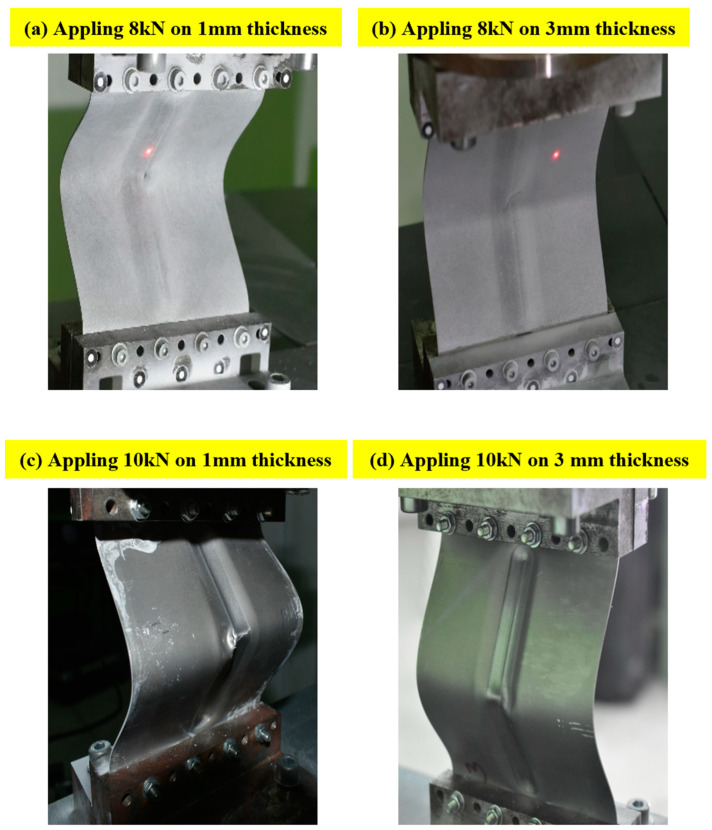
Aluminum alloy EN AW-2024-T3 sheets under compression load in different conditions.

**Figure 9 materials-17-01730-f009:**
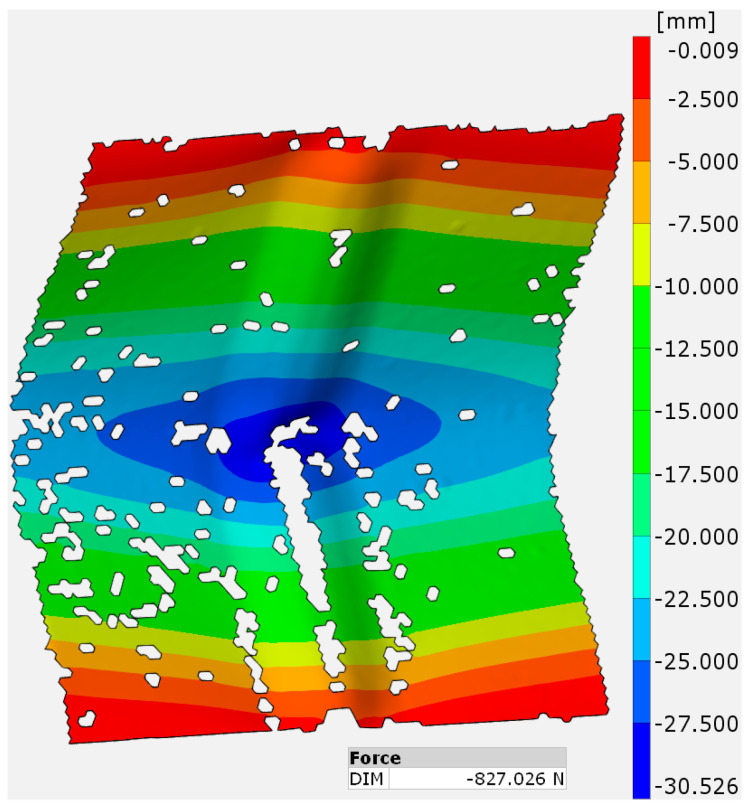
Deformation of aluminum 2024-T3 sheets just after destruction.

**Figure 10 materials-17-01730-f010:**
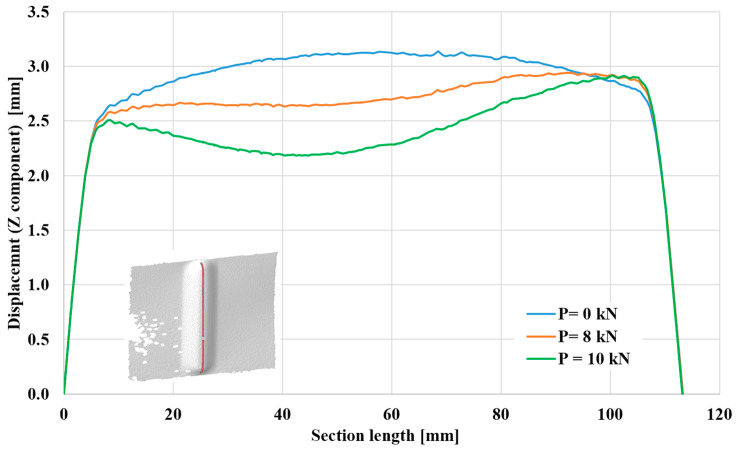
Deformation along section length of 1 mm thick aluminum alloy EN AW-2024-T3 sheets.

**Figure 11 materials-17-01730-f011:**
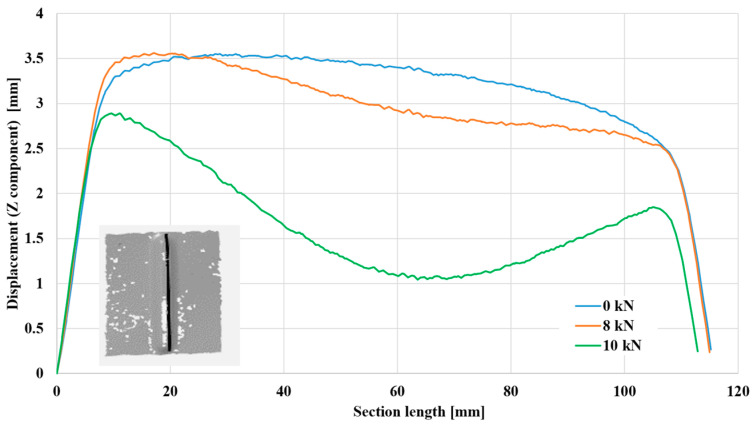
Deformation along section length of 3 mm thick aluminum alloy EN AW-2024-T3 sheets.

**Table 1 materials-17-01730-t001:** Chemical composition of EN AW-2024-T3 aluminum alloy sheets in wt.% [32].

Cu	Mg	Fe	Si	Mn	Zn	Ti	Cr	Others	Al
3.8–4.9	1.2–1.8	0.50	0.50	0.3–0.9	0.25	0.15	0.10	0.05–0.15	rest

**Table 2 materials-17-01730-t002:** Main mechanical properties of the EN AW-2024-T3 aluminum alloy sheet used.

Property	Value
Yield Stress (MPa)	332 ± 2.2
Ultimate Tensile Stress (MPa)	467 ± 1.9
Young Modulus (GPa)	71.6 ± 0.9
Poisson Ratio	0.33 ± 0.04

## Data Availability

Data are contained within the article.

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
