# Peer review of "Experimental Investigation of Load-Bearing Capacity in EN AW-2024-T3 Aluminum Alloy Sheets Strengthened by SPIF-Fabricated Stiffening Rib"

_materials, 2024, doi:10.3390/ma17081730_

Round 1

Reviewer 1 Report

Comments and Suggestions for Authors

This article describes the experimental investigation of load-bearing capacity in EN AW-2024-T3 aluminum alloy sheets strengthened by Single Point Incremental Forming (SPIF)-fabricated stiffening rib.

To improve the manuscript, the authors should consider the following modifications:

(1) The authors presented the schematic representation of symmetric SPIF in Figure 1. The authors should include the schematic representation of asymmetric SPIF for comparison purposes.

(2) The authors presented the main mechanical properties of the used EN AW-2024-T3 aluminum alloy sheet in Table 2.  The authors should include the standard deviations of the Yield Stress (MPa), Ultimate Tensile Stress (MPa), Young Modulus (GPa), and Poisson Ratio for reproducibility purposes.

(3) The authors presented the deformation along section lengths of 1 mm and 3 mm thicknesses of aluminum alloy EN AW-2024-T3 sheets in Figures 9 and 10, respectively. However, the authors should study the mathematical modeling, simulation, and optimization of the deformation behavior of the aluminum alloy EN AW-2024-T3 sheets to gain a better understanding of the deformation along section length.

The submitted manuscript has significant scientific insights and the experimental data support the conclusions. However, the present submission requires major revisions before being considered for publication in the Special Issue: Forming Technologies and Mechanical Properties of Advanced Materials - 2nd Volume in the esteemed Materials in its current condition. I hope the authors will find my comments helpful.

Author Response

Reviewer #1

Quality of English Language

( ) I am not qualified to assess the quality of English in this paper
( ) English very difficult to understand/incomprehensible
( ) Extensive editing of English language required
( ) Moderate editing of English language required
( ) Minor editing of English language required
(x) English language fine. No issues detected

Yes

Can be improved

Must be improved

Not applicable

Does the introduction provide sufficient background and include all relevant references?

( )

(x)

( )

( )

Are all the cited references relevant to the research?

(x)

( )

( )

( )

Is the research design appropriate?

( )

(x)

( )

( )

Are the methods adequately described?

( )

(x)

( )

( )

Are the results clearly presented?

( )

(x)

( )

( )

Are the conclusions supported by the results?

(x)

( )

( )

( )

Thank you for your positive feedback.

In the revised version, we improved the introduction, research design, method description, and clarity of results. We hope that will be compatible with your view.

Comments and Suggestions for Authors

This article describes the experimental investigation of load-bearing capacity in EN AW-2024-T3 aluminum alloy sheets strengthened by Single Point Incremental Forming (SPIF)-fabricated stiffening rib.

To improve the manuscript, the authors should consider the following modifications:

  • The authors presented the schematic representation of symmetric SPIF in Figure 1. The authors should include the schematic representation of asymmetric SPIF for comparison purposes.

Ans: Thank you very much for your valuable notice. The below text is added to the revised manuscript.

It is worth noting that SPIF can be symmetric or asymmetric, which is determined by the nature of the desired shaping process applications. The first type is preferred for all simple geometric shapes with better distribution of stresses, while the other type (often called asymmetric incremental sheet forming (AISF)) can form more complex shapes but is more wasteful of materials [14].

[14] Jeswiet, J., Micari, F., Hirt, G., Bramley, A., Duflou, J., & Allwood, J. (2005). Asymmetric Single Point Incremental Forming of Sheet Metal. CIRP Annals, 54(2), 88–114. doi:10.1016/s0007-8506(07)60021-3

  • The authors presented the main mechanical properties of the used EN AW-2024-T3 aluminum alloy sheet in Table 2.  The authors should include the standard deviations of the Yield Stress (MPa), Ultimate Tensile Stress (MPa), Young Modulus (GPa), and Poisson Ratio for reproducibility purposes.

Thank you for your valuable feedback. We appreciate your insightful suggestion regarding the inclusion of standard deviations for the mechanical properties of the EN AW-2024-T3 aluminum alloy sheet presented in Table 2. Recognizing the importance of reproducibility in research, we acknowledge the significance of providing comprehensive data to enhance the transparency and reliability of our findings, Table 2 is modified in revised version.

Table 2. Main mechanical properties of used EN AW-2024-T3 aluminum alloy sheet.

Properties

Value

Yield Stress (MPa)

332 ± 2.2

Ultimate Tensile Stress (MPa)

467 ± 1.9

Young Modulus (GPa)

71.6 ± 0.9

Poisson Ratio

0.33 ± 0.04

  • The authors presented the deformation along section lengths of 1 mm and 3 mm thicknesses of aluminum alloy EN AW-2024-T3 sheets in Figures 9 and 10, respectively. However, the authors should study the mathematical modeling, simulation, and optimization of the deformation behavior of the aluminum alloy EN AW-2024-T3 sheets to gain a better understanding of the deformation along section length.

Ans: We sincerely appreciate your valuable observation regarding the deformation behavior of the aluminum alloy EN AW-2024-T3 sheets presented in Figures 9 and 10. Your suggestion regarding the study of mathematical modeling, simulation, and optimization of deformation behavior is indeed pertinent and aligns with our research interests.

The research team is fully compatible with your valuable perceptions and what you have already indicated is being worked on in another parallel study of ours. The parallel research endeavor focuses on several key aspects, including the mathematical modeling, simulation, and optimization of deformation behavior along section lengths.

Specifically, our aim is in current study to investigate the deformation characteristics under varying pressure loads, particularly focusing on axial load shortening curves in EN AW-2024-T3 aluminum sheets reinforced with SPIF. Furthermore, our study expands its scope to explore the effect of deformation on the strengthening of aluminum alloy sheets through the utilization of an advanced digital image correlation system.

Once again, we express our gratitude for your insightful feedback, which has provided valuable direction for our ongoing and future research efforts.

The submitted manuscript has significant scientific insights and the experimental data support the conclusions. However, the present submission requires major revisions before being considered for publication in the Special Issue: Forming Technologies and Mechanical Properties of Advanced Materials - 2nd Volume in the esteemed Materials in its current condition. I hope the authors will find my comments helpful.

Thank you for your constructive feedback on our submitted manuscript. We appreciate your acknowledgment of the significant scientific insights and the support provided by the experimental data for our conclusions. Your recognition of the value of our work is encouraging.

We take your comments seriously and understand the need for major revisions.

We are committed to addressing your suggestions comprehensively, and we hope the revised version of the manuscript will ensure that the revised manuscript answers your inquiries and meets the high standards expected for publication in your esteemed journal.

Again, thank you for your thoughtful comments and for considering our manuscript for publication in the current special issue.

Reviewer 2 Report

Comments and Suggestions for Authors

This manuscript investigated the effect of stiffening rib on a metal sheet. Two different dimensions were studied and compared with compressive loading. The strain of the metal sheets were got from DIC. Overall, it has some merit to the industry. But the description needs to be improved to make it clear to the readers. See detailed comments below:

1. Line 40. This is the first time the idea of stiffening ribs occurs in this manuscript, so, a detailed description of it should be provided. What is stiffening ribs? What are the practical conditions. A figure with and without stiffening ribs is highly suggested to show this idea.

2. Line 113-120. The innovation of this work should be reconsidered. What are the properties others haven't investigated, what are the challenges, etc. Currently, the innovation, motivation, as well as the content of this work are not clear.

3. Line 169. What is Axial–load shortening Curve? The explanation should be provided at the beginning of this section.

4. Line 181. The experimental setup should be described more clearly. How was the samples clamped? What is the loading rate? An image is highly suggested to show the experimental setup.

5. Line 184-185. Are you describing a failed experiment? If so, I suggest dropping it off and only leave the successful ones. 

6. Line 271. Same with comment 3, What does it mean by " Deformation along section length"? What is the section length? and what kind of deformation are you measuring? A figure is suggested.

Author Response

Reviewer #2

Quality of English Language

( ) I am not qualified to assess the quality of English in this paper
( ) English very difficult to understand/incomprehensible
( ) Extensive editing of English language required
( ) Moderate editing of English language required
( ) Minor editing of English language required
(x) English language fine. No issues detected

Yes

Can be improved

Must be improved

Not applicable

Does the introduction provide sufficient background and include all relevant references?

( )

(x)

( )

( )

Are all the cited references relevant to the research?

(x)

( )

( )

( )

Is the research design appropriate?

(x)

( )

( )

( )

Are the methods adequately described?

( )

( )

(x)

( )

Are the results clearly presented?

( )

(x)

( )

( )

Are the conclusions supported by the results?

(x)

( )

( )

( )

Thank you for your positive feedback.

In the revised version, we improved the method description, and clarity of results. We hope that will be compatible with your view.

Comments and Suggestions for Authors

This manuscript investigated the effect of stiffening rib on a metal sheet. Two different dimensions were studied and compared with compressive loading. The strain of the metal sheets were got from DIC. Overall, it has some merit to the industry. But the description needs to be improved to make it clear to the readers. See detailed comments below:

ANS: Thank you for your feedback on our manuscript investigating the effect of stiffening ribs on metal sheets under compressive loading. We will carefully address your detailed comments and strive to improve the clarity of the description for better reader understanding.

  1. Line 40. This is the first time the idea of stiffening ribs occurs in this manuscript, so, a detailed description of it should be provided. What is stiffening ribs? What are the practical conditions. A figure with and without stiffening ribs is highly suggested to show this idea.

ANS: Thank you for your valuable feedback. Our manuscript acknowledges the need for a more detailed description of stiffening ribs. In the revised version, we provided a comprehensive explanation of stiffening ribs, including types, practical conditions, and potential applications, as below:

In engineering applications, stiffening ribs stand out as structural strengthening elements, are often added to metal panels and sheets, and have recently been extended to include composite materials and polymers to enhance their rigidity, strength, and resistance to deformation under load. Usually, stiffening ribs are raised or protruding features that run along the surface of the sheets.

Depending on their shape, many types of stiffening ribs are available, such as straight ribs, curved ribs, perforated ribs, grid ribs, tapered ribs, embossed ribs, and composite ribs[6].

In general, using stiffening ribs is a crucial part of improving the panels' structural integrity [7].

  1. Line 113-120. The innovation of this work should be reconsidered. What are the properties others haven't investigated, what are the challenges, etc. Currently, the innovation, motivation, as well as the content of this work are not clear.

ANS: Thank you for your thoughtful feedback regarding the innovation and motivation of our work. In response to your comments, we have revised the text in lines 113–120 to provide greater clarity on the innovation and motivation of our work.

Revised text:

"It is evident that there is a continued need for additional research in the field of SPIF-strengthened materials, particularly aluminum alloys such as EN AW-2024-T3 aluminum sheets. The primary objective of the current study is to address this gap and elucidate the axial-load shortening curves in EN AW-2024-T3 aluminum sheets strengthened by SPIF. Additionally, it analyzes the deformation behavior of the vertical section under various compressive loads. It also assesses how load-bearing capacity affects the thicknesses of EN AW-2024-T3 aluminum alloy sheets and investigates the deformation effect on the strengthened aluminum alloy sheets using an advanced digital image correlation (DIC) system. In summary, the study aims to integrate these comprehensive analyses to develop a new vision for understanding the  behavior of SPIF-strengthening aluminum alloys."

  1. Line 169. What is Axial–load shortening Curve? The explanation should be provided at the beginning of this section.

ANS: Thank you for your valuable notice. The first paragraph of section (3.1. Axial–load shortening Curve) in revised version will be as below.

The load versus end-shortening curves characterize the average stress-strain response of longitudinally stiffened panels subjected to compressive loading [35]. The axial-load shortening curves provide a valuable understanding of the material's de-formation characteristics, elucidating how it responds to varying levels of applied loads [36]. Usually, the x-axis represents the axial displacement or shortening of the material. It represents the deformation or compression experienced by the material under the applied axial load during compression testing, represented by the y-axis. In general, the axial-load shortening curves serve as informative tools for assessing axial stiffness, yield characteristics, and failure mechanisms in sheets while also facilitating the observation of local buckling behavior in aluminum sheets.

In aluminum applications, particularly in the aerospace industry, understanding the behavior of SPIF-strengthened aluminum sheets toward different compressive loads is crucial to predicting potential failure and structural integrity.

  1. Line 181. The experimental setup should be described more clearly. How was the samples clamped? What is the loading rate? An image is highly suggested to show the experimental setup.

ANS: Thank you for your valuable notice, required sample preparation and clamped is described with add addition figure 3 in page 5. The revised text is below:

When the aluminum alloy EN AW-2024-T3 sheets were rib‐stiffened, both the 1 mm thickness and the 3 mm thickness were examined in a compression test using the Z100 testing machine (Zwick/Roell, Ulm, Germany) for two ultimate loads of 8 kN and 10 kN, respectively.  Using a Z100 machine based on the ASTM E9/E9M standard, all EN AW-2024-T3 aluminum samples strengthened by SPIF were compressed using two compressive loads (8 kN and 10 kN) at room temperature. The compression testing machine securely mounted the specimens, aligned them perpendicular to the loading direction, and well-fixed them to prevent slippage during the test. Then, the compressive force was applied to the specimen at the test speed (or initial strain rate of 1 mm/min) until failure occurred or until the desired deformation was reached as shown in Figure 3.

Figure 3. A compression machine with specimen fixation (after test).

  1. Line 184-185. Are you describing a failed experiment? If so, I suggest dropping it off and only leave the successful ones. 

ANS: Thank you for this essential note. In fact, it was expected that the first attempt would be deleted, but the research team thinks that it will help compare the results. We applied a test load to the sample to determine the scope of the sample’s initial bearing. The sample remained steadfast during this attempt and did not develop a buckling. In the second attempt, a higher load was applied to the same sample, and the expected deformation and buckling occurred. For the 3 mm thickness, there was only one attempt where the maximum load became known. All results were recorded in Figure 5 for comparison and to give the impression that the sample was preloaded, which is better for comparison with other results.

  1. Line 271. Same with comment 3, What does it mean by " Deformation along section length"? What is the section length? and what kind of deformation are you measuring? A figure is suggested.

ANS: Thank you for your valuable notice.

The strengthening section area is 110 mm long and 5 mm wide from the specimen. The section length means the length of deformation from the specimen's lower point to the upper point. So, this part investigates the deformation at each point along this length using high-resolution cameras based on digital image correlation (DIC).

Round 2

Reviewer 1 Report

Comments and Suggestions for Authors

Dear Authors: Many thanks for your sincere efforts in improving your manuscript. The revised article is highly satisfactory and merits acceptance for publication in the Special Issue: Forming Technologies and Mechanical Properties of Advanced Materials - 2nd Volume in the Materials.